# Genetic Diversity of *Colletotrichum* spp. Causing Grape Anthracnose in Zhejiang, China

Boyang Ye [1], Jingqun Zhang [1], Xiangyang Chen [1], Wenfei Xiao [2], Jianyan Wu [1], Hong Yu [2],* and Chuanqing Zhang [1],*

[1] College of Advanced Agricultural Sciences, Zhejiang A&F University, Hangzhou 311300, China
[2] Research Institute for the Agriculture Science of Hangzhou, Hangzhou 310013, China
* Correspondence: yuh5060@gmail.com (H.Y.); cqzhang@zafu.edu.cn (C.Z.)

**Abstract:** Anthracnose is a fungal disease that seriously threatens grape production and quality. Multiple *Colletotrichum* species are detected in anthracnose grapes in vineyards. In this study, diseased grapes were collected in four counties in Zhejiang, and 43 *Colletotrichum* isolates were obtained. Multigenes (ITS, *TUB2*, *ACT*, *CHS-I*, and *GAPDH*) and morphological characteristic analyses showed that *C. fructicola* (40 isolates, 93%), *C. aenigma* (two isolates, 4.7%), and *C. pseudoacutatum* (one isolate, 2.3%) were the cause of grape anthracnose in Zhejiang. Among the three *Colletotrichum* species, *C. fructicola* was the prevalent and dominant species in all sampled counties; *C. pseudoacutatum* was first identified as the pathogen responsible for grape anthracnose. There were significant differences in the sporulation among the three *Colletotrichum* species, as well as in the spore germination. Pathogenicity testing showed that all species can infect grapes, resulting in anthracnose. On the other hand, the virulence of species was varied and may be associated with their spore germination. This is the first study to characterize the *Colletotrichum* species causing grape anthracnose in Zhejiang Province and reveal that *C. fructicola* is the dominant species. The determination of *Colletotrichum* species associated with grape anthracnose may contribute to the study of epidemiology and development of an efficient strategy for controlling anthracnose in the vineyards.

**Keywords:** grape anthracnose; *Colletotrichum* species; multi-gene analysis; *C. fructicola*; *C. pseudoacutatum*; spore germination; pathogenicity





## 1. Introduction

Grape berries are popular among customers, both as a fresh-eating food and as a material for juice and wine production. In addition, the berries are rich in various phenolic compounds, flavonoids and stilbenes, which are of benefit to human health and may help prevent several diseases and are beneficial to health [1,2]. Grape is one of the most extensively planted fruit trees worldwide, with planting ranging from temperate to tropical regions. According to the Food and Agriculture Organization (FAO), the grape-growing area is increasing by above 2% per year. China is the largest grape producer and consumer, with a total planting area of about 700,000 km$^2$ and a production of about 1.3 million tons in 2018 [3]. Grape anthracnose or ripen rot is a serious disease; its occurrence at flowering and at harvest causes grape yield loss [4–6]. Furthermore, the disease also results in a bitter taste to the fruit and wine because of the increased chemical compositions (i.e., volatile acidity, residual sugar, glycerol, gluconic acid, and malic acid) in diseased fruit [7,8].

Grape anthracnose occurs in most vineyards worldwide, and the disease is caused by *Colletotrichum* spp. or by *Elsinoe ampelina* [9,10]. The symptoms induced by *E. ampelina* differ from those caused by *Colletotrichum* spp.. For instance, berries infected by *E. ampelina* first form dark-brown spots, then turn into sunken lesions with a grey-white center and dark-brown to purple-brown margins [10]; berries infected by *Colletotrichum* spp. form circular, reddish-brown spots that enlarge in concentric lesions, and acervuli with orange spore masses are observed in the spots [4–6,11].

*Colletotrichum* spp. is ranked eighth on the top 10 fungal plant pathogen list and causes anthracnose spots, wilting of plant parts, and post-harvest rot [12]. *Colletotrichum* spp. are important post-harvest pathogens because of latent infections: fruit are infected at the young stage, but do not show any symptoms until after the fruit have been stored or been marketed; this characteristic can cause up to 100% loss at the store stage [13]. When *Colletotrichum* spp. infects grapes, it causes grape anthracnose or ripen rot [9,14]. Latent infection is also observed when grapes are infected from the young green stage to the ripe stage, but do not show symptoms until ripening [4,15].

*Colletotrichum* spp. infects grapevine leaves, stems, and especially fruit [6,8]. It was recognized that *C. gloeosporioides* species complex and *C. acutatum* species complex are responsible for grape ripe rot [4,5,8,11,14–16]. Several studies have shown that *C. gloeosporioides* is sensitive to benomyl and that *C. acutatum* is generally moderately resistant to benzimidazole fungicides [4,11]. In addition, the epidemiology of *C. gloeosporioides* differs from *C. acutatum* [17,18]. Therefore, a precise determination of *Colletotrichum* species on the host is important for controlling *Colletotrichum* diseases in the fields.

In the past, morphological characterizations (i.e., colony features, size and shape of conidia and appressoria, optimal temperature, growth rate) were used for the identification of *Colletotrichum* species [19–21]. However, because of the stability of morphological traits influenced by cultivation environments and the existence of intermediate forms, differentiation of *Colletotrichum* species based on morphological characters is not entirely reliable enough to distinguish between *Colletotrichum* species. Molecular techniques, such as polymerase chain reaction (PCR), real-time PCR, restriction fragment length polymerase (RFLP), and simple sequence repeat (SSR) markers have been developed to detect pathogens [15,22]. DNA markers based on ITS regions are always used to detect fungi, and the multi-loci phylogenetic analysis method was used to distinguish *Colletotrichum* at the species level [22–24]. Combining microbiological methods with molecular phylogeny led to a better understanding and identification of *Colletotrichum* species, which has been used for the identification of *Colletotrichum* species on many hosts [25–27].

Zhejiang Province is one of the major grape-planting areas in southern China, and the grape industry is a characteristic agriculture for creating economic effectiveness for farmers in the region [3,28]. In Zhejiang, August and September are relatively warm summer months with the highest precipitation, which coincide with the final stages of grape ripening [28]. As we know, heat and wetness favor grape anthracnose epidemics. To provide basic knowledge of grape anthracnose and help control this fungal disease in Zhejiang, this study first investigated the genetic diversity of grape anthracnose and clarified the distribution of the *Colletotrichum* species in Zhejiang. Secondly, the biological characteristics and pathogenicity of each *Colletotrichum* species were determined.

## 2. Materials and Methods

### 2.1. Fungus Isolation

Grape fruit (*Vitis vinifera* 'Kyoho') with characteristic anthracnose symptoms were sampled from distinct vineyards in four counties in Zhejiang Province. We randomly selected six, four, five, and four vineyards in Hangzhou, Taizhou, Jinhua, and Shaoxing, respectively, for collecting samples. A total of 98 diseased fruit were collected, including 27 fruit from Hangzhou, 18 fruit from Taizhou, 27 fruit from Jinhua, and 26 fruit from Shaoxing, respectively. In the laboratory, diseased fruit were incubated in a growth chamber at 25 °C with a photoperiod of 12 h and 95% relative humidity until orange mucilage containing conidia formed on the lesions. The monospore culture method was then established as described in Arzanlou et al. [29]. In brief, conidia were picked up from diseased samples using a sterile needle, and transferred into 10 mL of sterile distilled water. Next, 100 μL of the conidia suspension was spread evenly on 2% water agar plates containing 50 mg/L streptomycin sulfate and incubated overnight at 25 °C. Then, the plates were checked under a light microscope (ZEISS Axio Scope. A1, Jena, Germany) and germinated conidia were transferred to a potato dextrose agar (PDA) plate. Ten conidia from each plate were

cultured separately and the strains with different colony morphology were maintained on PDA slants at 4 °C.

### 2.2. Molecular Characterization

The mycelia of each field *Colletotrichum* isolates were harvested after 6 days of cultivation at 25 °C. Mycelia were momentarily frozen in liquid nitrogen and ground into a fine powder using a sterile mortar and pestle. The powdered mycelia were dissolved in buffer digestion (Sangon Biotech, Shanghai, China), then genomic DNA was extracted using the Rapid Fungi Genomic DNA Isolation Kit (Sangon Biotech) according to the manufacturer's instructions. The DNA concentration was determined by spectrophotometry and diluted with distilled water for further DNA amplification reactions. Internal transcribed spacer (ITS), partial beta tubulin 2 (*TUB2*), acting (*ACT*), chitin synthase (*CHS-I*), and glyceraldehyde-3-phosphate dehydrogenase (*GAPDH*) genes were amplified using the primer pairs ITS-1F/ITS4 [30], T1/T2 [31], ACT-512F/ACT-783R [32], CHS-79F/CHS-354R [32], and GDF/GDR [33], respectively.

Each 50 μL of gene amplification reaction mixture contained 2× Taq PCR Master Mix (25 μL, Sangon Biotech), template DNA (1 μL, 20–50 ng), 10 μM forward primer and reverse primer (2 μL, each primer). PCR conditions for ITS, *ACT*, *CHS-I* and *GAPDH* were performed as described by Chen et al. [34]. The PCR condition for *TUB2* was conducted with denaturation at 94 °C for 4 min, followed by 30 cycles of 94 °C for 30 s, 55 °C for 30 s, 72 °C for 45 s, and further extension at 72 °C for 10 min. The amplicons were separated by electrophoresis and visualized by staining with ethidium bromide (EB). The single band with expected size was cut and recovered using the SanPrep Column PCR Product Purification Kit (Sangon Biotech), then the purified products were sequenced by Sangon Biotech.

### 2.3. Multi-Locus Phylogenetic Analysis

Raw sequence chromatograms were manually examined, and the sequences for each fragment were assembled by primer pair of each gene using DNAStar v. 5.0 software. The assembled sequences of ITS, *TUB2*, *ACT*, *CHS-I*, and *GAPDH* were shared in the GenBank database of the National Center for Biotechnology Information (Table S1) and also were blasted against the GenBank database, respectively. *Colletotrichum* isolates were then first identified at the species complex level. Related gene sequences (ITS, *TUB2*, *ACT*, *CHS-I*, and *GAPDH*) of *Colletotrichum* species from previous publications were downloaded from GenBank [23,24]. The referenced standard strains are listed in Table S1. For each gene, sequences from the isolates belonging to the same species complex were compared using MAFFT v.7 following default parameter settings [35]. If necessary, sequences were manually adjusted in MEGA v. 6.06 [36]. The modified sequences were concatenated in Sequence Matrix 1.8. Modeltest3.7.win, Win-paup4b10-console, and Mrmodeltest2, as implemented in MrMTgui, and were used to estimate the best model of nucleotide substitution [37]. Bayesian inference (BI) was used to construct phylogenies using MrBayes v. 3.1.2 [38]. Six simultaneous Markov chains were run for 5,000,000 generations each until the standard deviation of split frequencies was below 0.01, and trees were sampled every 100th generation. The first 2000 trees, representing the burn-in phase of the analyses, were discarded, and the remaining 8000 trees were used to calculate the posterior probability (PP) in the majority rule consensus tree. Phylogenetic trees were drawn using TreeView [39]. The alignments and trees were deposited into TreeBase.

### 2.4. Morphological Characterization

Representative isolates of each *Colletotrichum* species were cultured on the PDA, malt extracts agar (MEA; Sigma Aldrich, UK) and oatmeal agar (OA; Sigma-Aldrich, Gillingham, UK), respectively. Their colonies characterization was performed following the procedures of Karimi et al. [40]. One fresh mycelium plug (5 mm in diameter) of *Colletotrichum* isolates was taken from the growing edge of 4-day-old cultures and was transferred onto the

center of PDA, MEA, and OA plates. After cultivation for 7 days at 25 °C in darkness, the colony morphology was recorded, and the cultures on PDA plates were continued until conidiomata formed. The morphology of conidiomata were recorded by a light microscope (Nikon SMZ25, Tokyo, Japan), then conidiomata was picked up and diluted in distilled water to obtain a conidia suspension with a density of $1 \times 10^6$ conidia·mL$^{-1}$. Fifty spores were selected randomly to measure the length and width under a Nikon Eclipse 80i microscope (Tokyo, Japan). Conidia suspension was also used to induce appressoria as described in Wu et al. [25], and the morphology and size of 30 appressoria were recorded.

### 2.5. Mycelial Growth, Sporulation and Conidial Germination of Colletotrichum spp.

Isolates chosen in the previous section were also used to determine the temperature effect on colony growth. As described previously [41], for the determination of temperature effects, a mycelial plug (5 mm in diameter) from the growing edge of 4-day-old cultures was transferred onto the center of PDA plate (90 mm in diameter), and each isolate was inoculated separately at 5, 10, 15, 17, 20, 25, 27, 30, 35, and 40 °C for 5 days. Each treatment was set up for three replicates and the experiment was performed three times. The optimum temperature for the mycelial growth was estimated based on the Gaussian process (least squares fit) for nonlinear regression in Graph Prism 8.0 [42].

Additionally, sporulation and conidial germination of the selected isolates were also determined in this study. For conidia production, 10 fresh mycelial plugs were transferred to a 250-mL flask containing 100 mL potato dextrose (PD) liquid medium and were shaken at 160 rpm for 5 days at 25 °C. The conidial suspension of each isolate was filtered through three layers of muslin cloth, a 10-μL mixture was dropped on a hemocytometer and covered with a coverslip. The conidia production of each isolate was counted under a ZEISS Axio Scope A1 microscope. The mean spore numbers from ten biological replicates were used to represent sporulation capacity.

Aliquot of 6 μL suspensions with $1 \times 10^6$ spores mL$^{-1}$ were dropped on plastic unbreakable cover slip (size 22 × 22 mm) (Thermos Fisher Scientific, Waltham, MA, USA), and then incubated at 25 °C in darkness. After 16 h incubation, germination was quantified at five sites by counting 200 conidia in total, and the germination percentage was calculated. When a germ tube reached at least half the length of the conidium, it was scored as germinated [17]. Five biological replicates were analyzed for each isolate.

### 2.6. Pathogenicity Testing

Berries of 'Kyoho' were used for evaluating the pathogenicity of *Colletotrichum* spp. Healthy grapes with a uniform size and no visible surface symptoms were selected for the pathogenicity test. Inoculation was performed according to Yan et al. [9], with some modifications: grapes were surface disinfected with 2% NaOCl for 2 min followed by three rinses in sterile water, and then air-dried on sterilized filter paper. The grapes were inoculated by the wound/drop incubation method. A sterilized insect pin (0.5 mm in diameter) was used to create superficial wounds in the epidermis on the detached fruit, and an aliquot of 6 μL of conidia suspension with $1 \times 10^6$ spores mL$^{-1}$ was dropped on the wound. Inoculated grapes were incubated in a growth chamber at 25 °C with a photoperiod of 12 h and 95% relative humidity. Grapes treated with 6 μL of sterile water on the wound were included as a control. The experiment was performed three times, using 10 grapes per treatment per assay. The pathogens were re-isolated from the resulting lesions by tissue isolation method described above.

After 2 days, incidence in grape fruit was counted, and after 5 days, disease severity in grape fruit was assessed as the percentage of fruit area with anthracnose symptoms as previously described [43]. The rating scale for fruit was classified as follows: (0) no symptoms; (1) 0.1–5.0% of the fruit surface affected; (3) 5.1–15.0%; (5) 15.1–30.0%; (7) 30.1–50.0%; (9) 50.1–100.0% or rotten fruit. The lesion area was used to calculate the disease indicated as follows: Disease index = $[(n_0 \times 0 + n_1 \times 1 + n_3 \times 3 + n_5 \times 5 + n_7 \times 7 + n_9 \times 9)/(9 \times N)] \times 100$,

where *n* represents the number of fruit in each class (0, 1, 3, 5, 7, 9) and *N* represents the total number of fruit.

*2.7. Statistical Analysis*

The mycelial growth, sporulation, spore germination, and virulence data were analyzed using IBM SPSS Statistics 29.0 by Fisher's least significant difference (*LSD*) with a one-way ANOVA at *p* = 0.05.

## 3. Results

*3.1. Colletotrichum Isolates Associated with Grape Anthracnose*

Grape berries showing anthracnose symptoms were sampled in vineyards from four different geographical regions in Zhejiang Province. In total, 43 of the diseased grapes were used for monospore isolation, and finally 43 fungal isolates were recovered based on primary colony morphology, including 13 isolates obtained from Hangzhou, 14 isolates obtained from Shaoxing, 11 isolates obtained from Jinhua, and 5 isolates obtained from Taizhou.

*3.2. Multi-Locus Phylogenetic Analyses*

All 43 field *Colletotrichum* isolates were grouped into three *Colletotrichum* species. In the phylogenetic tree constructed for the *C. gloeosporioides* species complex, 42 field isolates and 26 reference isolates (Table S1), including *C. boninense* (CBS: 123755) as an outgroup, were subjected to multi-locus phylogenetic analyses with concatenated ITS, *TUB2*, *ACT*, *CHS-I*, and *GAPDH* sequences. The concatenated alignment of ITS, *TUB2*, *ACT*, *CHS-I*, and *GAPDH* contained 1459 characteristics (gene boundaries, *ITS*: 1–370, *TUB2*: 371–690, *ACT*: 691–900, *CHS-I*: 901–1196, *GAPDH*: 1197–1459). The results showed that 42 isolates clustered together with two species: 40 isolates with *C. fructicola* and 2 isolates with *C. aenigma* (Figure 1).

The remaining one isolate HZJ-6 and 31 reference isolates, including *C. orchidophilum* (CBS 632.80) as an outgroup, were subjected to multi-locus phylogenetic analyses with concatenated ITS, *TUB2*, *ACT*, *CHS-I*, and *GAPDH* sequences from those belonging to *C. acutatum* species complex. Finally, isolate HZJ-6 did not belong to the *C. acutatum* species complex and was clustered with *C. pseudoacutatum* (Figure 2).

*3.3. Species Distribution*

Among the three *Colletotrichum* species, *C. fructicola* was detected in each sampled county (Figure 1). In total, 40 of the 43 isolates were grouped into *C. fructicola*, which indicated that *C. fructicola* was the main pathogen associated with grape anthracnose in Zhejiang Province. *C. aenigma* (two isolates) and *C. pseudoacutatum* (one isolate) both showed a lower frequency in Zhejiang Province and were only found in Hangzhou and Shaoxing, respectively. In Jinhua and Taizhou counties, only *C. fructicola* isolates were isolated from grape with anthracnose symptoms (Figure 3).

*3.4. Morphology Description*

Based on multi-locus sequence data, the 43 isolates were classified into three *Colletotrichum* species. Of these, *C. pseudoacutatum* was first recorded in grapes. The three *Colletotrichum* species were morphologically similar (Figures 4–6) and their morphology characteristics are described below.

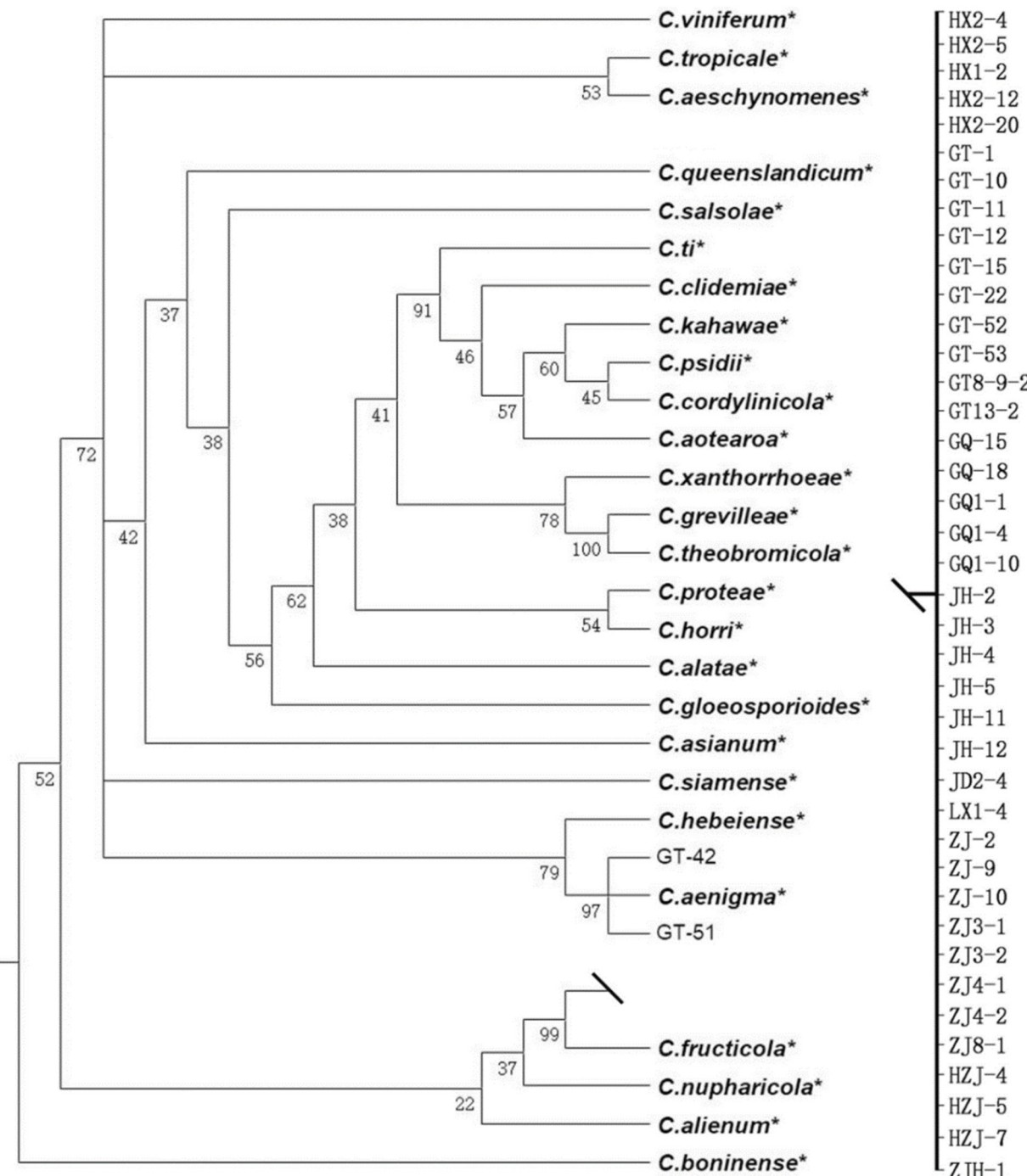

**Figure 1.** A Bayesian inference phylogenetic tree of the *Colletotrichum gloeosporioides* species complex. The tree was constructed with a Bayesian analysis of concatenated internal transcribed spacer (ITS), partial beta tubulin 2 (*TUB2*), acting (*ACT*), chitin synthase (*CHS-I*), and glyceraldehyde-3-phosphate dehydrogenase (*GAPDH*). The Markov chain was run for 5,000,000 generations, each locus having a separate model of DNA evaluation. *C. boninense* (CBS: 123755) was used as an outgroup. * The reference isolates.

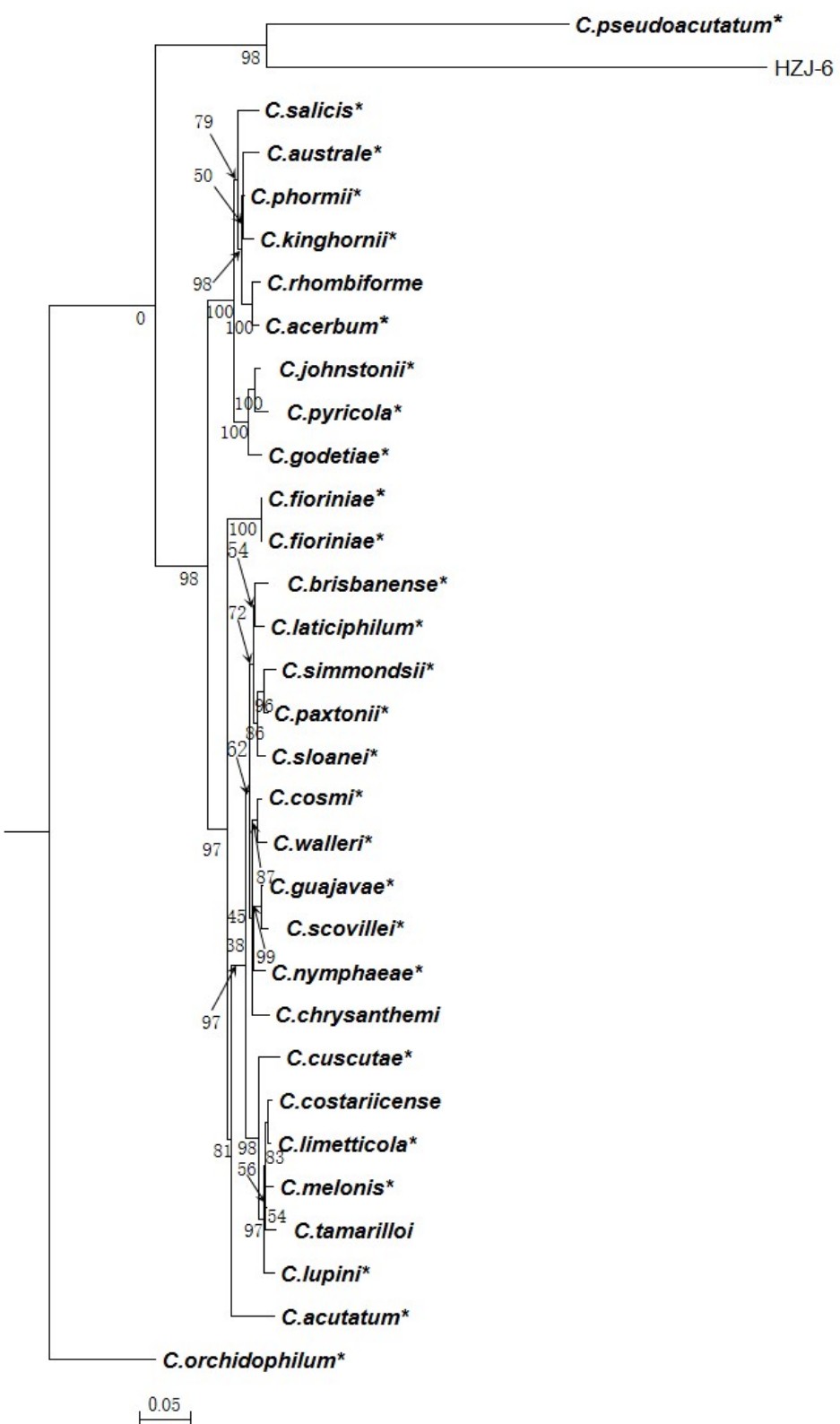

**Figure 2.** A Bayesian inference phylogenetic tree of the *C. acutatum* species complex. The tree was constructed with a Bayesian analysis of concatenated internal transcribed spacer (ITS), partial beta tubulin 2 (*TUB2*), acting (*ACT*), chitin synthase (*CHS-I*), and glyceraldehyde-3-phosphate dehydrogenase (*GAPDH*). The Markov chain was run for 5,000,000 generations, each locus having a separate model of DNA evaluation. *C. orchidophilum* (CBS 632.80) was used as an outgroup. The scale bar shows 0.05 expected changes per site. * The reference isolates.

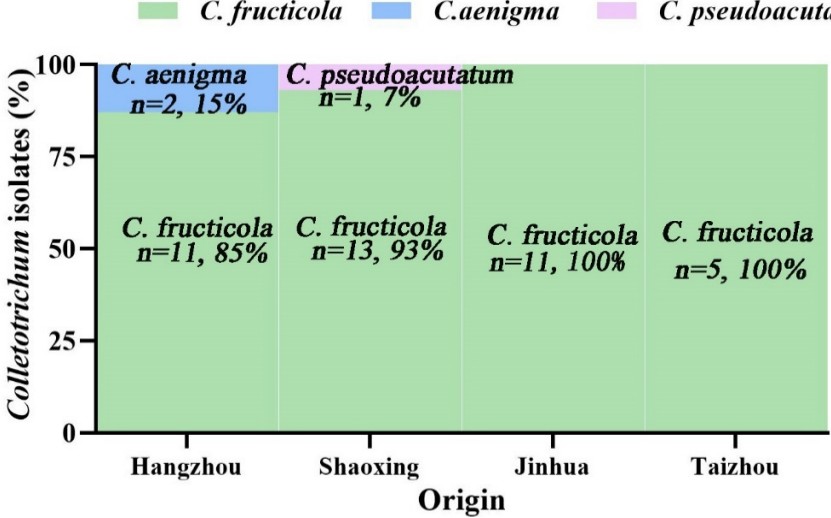

**Figure 3.** Distribution of *Colletotrichum* species in four grape-planting regions in Zhejiang, China. *n* = number of *Colletotrichum* isolates analyzed. 13 *Colletotrichum* isolates from Hangzhou contained *C. fructicola* (11 isolates, 85%) and *C. aenigma* (two isolates, 15%); 14 isolates from Shaoxing contained *C. fructicola* (13 isolates, 93%) and *C. pseudoacutatum* (one isolate, 7%); a further 11 isolates from Jinhua and 5 isolates from Taizhou all were *C. fructicola*.

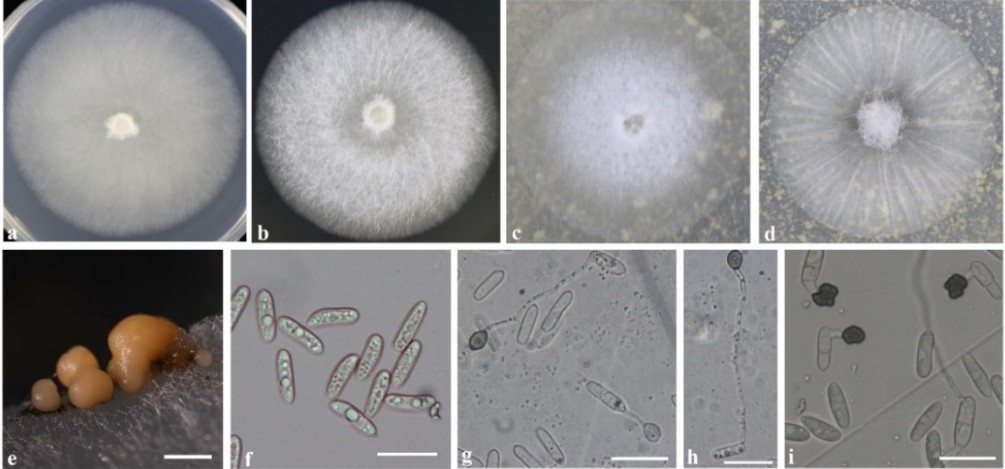

**Figure 4.** Culture characteristics and microscopic features of *C. fructicola*. (**a**,**b**) Seven-day-old of colonies on PDA and MEA medium, respectively. (**c**,**d**) Seven-day-old of colonies on OA medium. (**e**) Conidiomata on PDA medium. (**f**) Conidia. (**f**–**i**) Appressoria. Scale bars: (**e**) = 500 μm; (**f**–**i**) = 20 μm. (**a**–**c**,**e**–**h**): isolate GT-22; (**d**,**i**): isolate JH-2.

After 7 days of cultivation on a medium, colonies of *C. fructicola* on PDA were light gray, villous, and dense (Figure 4a); on MEA, they were white (Figure 4b), and on OA, they were white in the center with a transparent margin (Figure 4c) or grew radially with a thin, white mycelium (Figure 4d). Similar colony morphology was observed in *C. aenigma* (Figure 5a–c). On PDA, colonies of *C. pseudoacutatum* were atrovirens in the center with a gray margin; on MEA and OA, they grew with white mycelium and fluffy orange-yellow conidiomata distributed beneath the hyphae (Figure 6a–c). Conidia morphology was similar among the three species: conidia in mass orange, conidia hyaline, aseptate, smooth-walled, and cylindrical (Figures 4e,f, 5d,e and 6d,e). Few differences in appressoria morphology were observed among the three species. Appressoria of *C. fructicola* and *C. pseudoacutatum* were both ellipsoid or irregular (Figures 4g–i and 6f,g), and appressoria of *C. aenigma* were subglobose (Figure 6f,g). The conidia and appressoria sizes of the tested isolates are described in detail in Table 1.

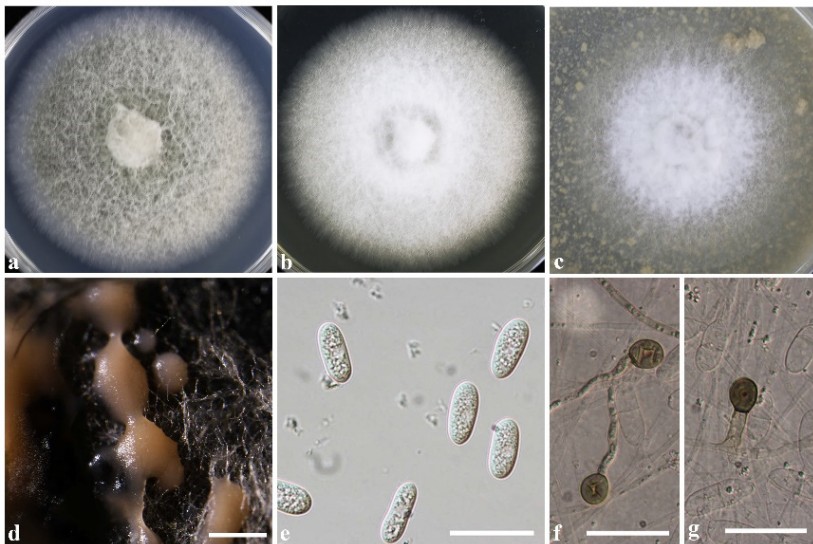

**Figure 5.** Culture characteristics and microscopic features of *C. aenigma* (isolate GT-42). (**a**–**c**) Seven-day-old of colonies on PDA, MEA, and OA medium, respectively. (**d**) Conidiomata. (**e**) Conidia. (**f**,**g**) Appressoria. Scale bars: (**b**) = 500 μm; (**e**–**g**) = 20 μm.

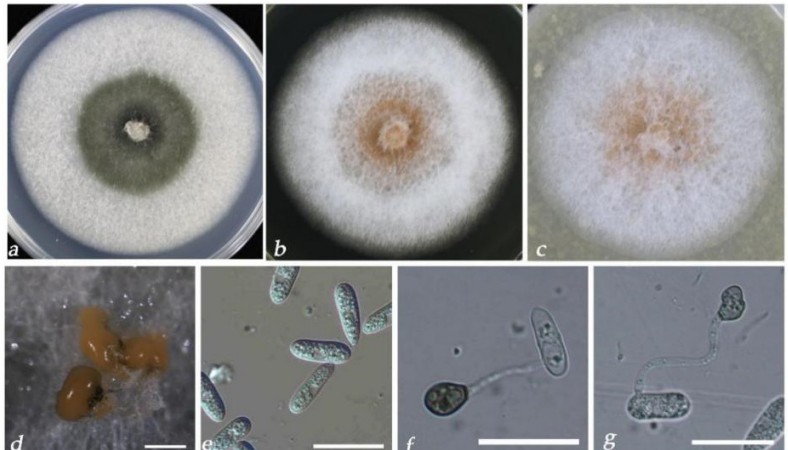

**Figure 6.** Culture characteristics and microscopic features of *C. pseudoacutatum* HZJ-6. (**a**–**c**) Seven-day-old of colonies on PDA, MEA, and OA medium, respectively. (**d**) Conidiomata. (**e**) Conidia. (**f**,**g**) Appressoria. Scale bars: (**d**) = 500 μm; (**e**,**f**) = 20 μm.

**Table 1.** Colony morphology and morphometry of *Colletotrichum* species recovered form grape.

| Species | Isolates | Colony Morphology [1] | | | Conidia | Appressorium |
| | | PDA [2] | MEA [3] | OA [4] | Size [5] (μm) | Size [5] (μm) |
|---|---|---|---|---|---|---|
| *C. fructicola* | GT-22 | R, C, G | R, C, W | R, C, W(c) to T(m) | 13.7–18.2 × 4.3–5.9 (16.9 ± 0.5) × (4.7 ± 0.5) | 6.4–8.5 × 4.8–7.3 (7.5 ± 0.7) × (5.7 ± 0.8) |
| | JH-2 | R, C, G | R, C, W | R, Ra, W | 13.1–18.6 × 4.2–5.9 (16.8 ± 0.5) × (4.9 ± 0.3) | 7.2–9.9 × 6.1–8.8 (8.0 ± 0.4) × (6.9 ± 0.5) |
| *C. aenigma* | GT-51 | R, C, GG | R, C, W | R, C, W(c) to T(m) | 13.2–17.1 × 4.1–5.6 (14.1 ± 0.4) × (5.1 ± 0.3) | 6.2–9.6 × 5.2–7.6 (6.9 ± 0.9) × (5.9 ± 0.6) |
| | GT-42 | R, C, GG | R, C, W | R, C, W(c) to T(m) | 13.1–17.8 × 3.8–5.7 (13.9 ± 0.5) × (4.5 ± 0.5) | 6.0–8.3 × 4.9–6.2 (6.8 ± 0.6) × (5.7 ± 0.5) |
| *C. pseudoacutatum* | HZJ-6 | R, C, A(c) to G(m) | R, C, O(c) to W(m) | R, C, O(c) to W(m) | 14.2–18.9 × 4.9–7.2 (16.6 ± 0.5) × (5.6 ± 0.4) | 7.5–10.1 × 5.2–7.2 (8.7 ± 0.9) × (5.4 ± 0.8) |

[1] Surface of colony: R, regular; C, circular; G, gray; GG, gray-green; A, atrovirens; W, white; O, orange; T, transparent; Ra, radiation; c, center; m, margin. [2] Potato dextrose agar. [3] Malt extracts agar. [4] Oatmeal agar. [5] Length × Width (data are mean ± standard deviation).

### 3.5. Mycelial Growth, Sporulation and Conidial Germination of Colletotrichum spp.

Five isolates selected for morphology observation were grown on PDA at 5 °C to 40 °C. The maximum colony diameter of *C. fructicola* GT-22, *C. fructicola* JH-2, *C. aenigma* GT-51, *C. aenigma* GT-42, and *C. pseudoacutatum* HZJ-6 after cultivation for 5 days on PDA was at 24.8 °C, 25.4 °C, 25.1 °C, 25.2 °C, and 25.0 °C, respectively (Figure 7a; Table S2). The temperature of 25 °C to 30 °C was suitable for the mycelial growth of *C. fructicola* and *C. pseudoacutatum*, and 25 °C to 27 °C was suitable for the mycelial growth of *C. aenigma* (Figure 7a, Tables S2 and S3).

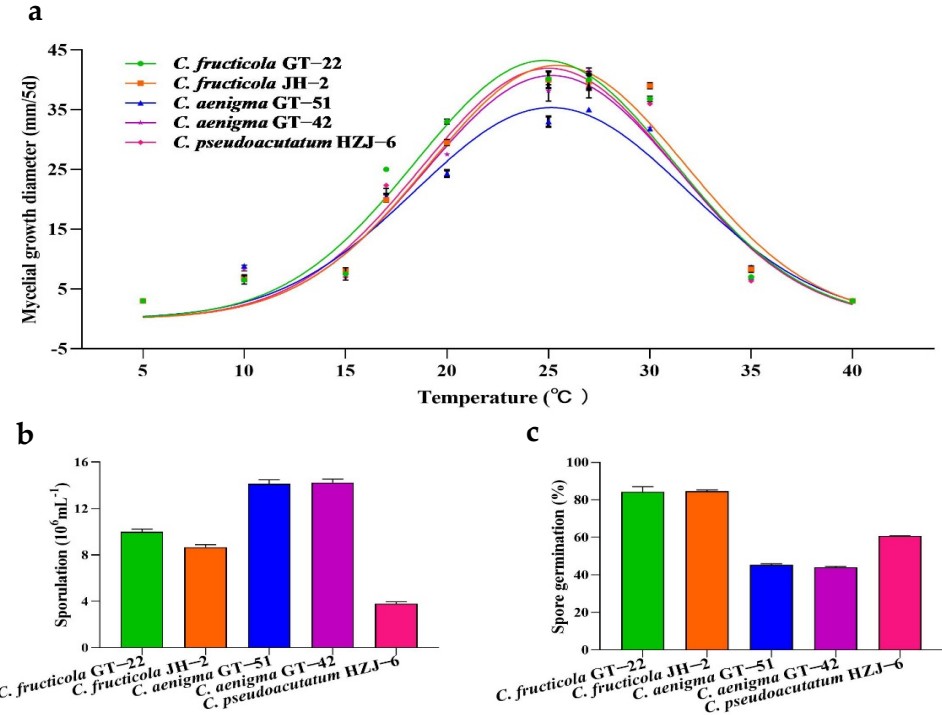

**Figure 7.** Biological characteristics of *Colletotrichum* species. (**a**) Colony diameter of *Colletotrichum* species at different temperatures. The colony diameter of each *Colletotrichum* isolate is the mean ± standard error of three independent experiments. Different symbols represent the mean colony diameter of different species at the tested temperatures. Gaussian process regression was used to estimate the optimum temperature for mycelial growth. Comparison of sporulation (**b**) and spore germination (**c**) of *Colletotrichum* species.

Based on the conidia production and conidia germination results (Figure 7b,c; Table 2), the ranking of species by sporulation under the same environment was as follows: *C. aenigma* GT-51 and GT-42 > *C. fructicola* GT-22 > *C. fructicola* JH-2 > *C. pseudoacutatum* HZJ-6. The ranking of species by spore germination was as follows: *C. fructicola* GT-22 and JH-2 > *C. aenigma* GT-51 and GT-42 > *C. pseudoacutatum* HZJ-6.

**Table 2.** Characterization of *Colletotrichum* species recovered from grapes.

| Species | Isolates | Sporulation [1] ($\times 10^6$ mL$^{-1}$) | Spore Germination [1] (%) | Disease Index [1] |
|---|---|---|---|---|
| *C. fructicola* | GT-22 | 10.0 ± 0.2 b | 84.2 ± 2.8 a | 44.8 ± 5.7 a |
| | JH-2 | 8.7 ± 0.2 c | 84.5 ± 0.8 a | 45.8 ± 5.9 a |
| *C. aenigma* | GT-51 | 14.1 ± 0.4 a | 45.3 ± 0.6 c | 14.7 ± 2.1 b |
| | GT-42 | 14.2 ± 0.3 a | 44.0 ± 0.5 c | 14.0 ± 3.9 b |
| *C. pseudoacutatum* | HZJ-6 | 3.8 ± 0.2 d | 60.7 ± 0.3 b | 28.0 ± 1.7 ab |

[1] Values shown are the means ± standard error with different lowercase letters indicating significant differences ($p < 0.05$) according to Fisher's least significant difference test.

*3.6. Pathogenicity Tests*

The pathogenicity test showed that the lesion area of grapes treated with sterile water on the wound was less than 0.1%, and significant differences in virulence ($p < 0.05$) were observed in three *Colletotrichum* species. Necrotic spots on grape fruit caused by *C. fructicola* isolates and *C. pseudoacutatum* isolate were visible at 2 days post inoculation (dpi). At 5 dpi, the disease index caused by *C. fructicola* GT-22 and JH-2 was 44.7 and 45.8, respectively, showing the greatest disease severity (Table 2). In contrast, *C. aenigma* species showed significantly weaker pathogenicity with a mean disease index of 14.4 (Table 2).

## 4. Discussion

Grape anthracnose is one of the important fungal diseases in vineyards and is epidemic worldwide. As previous research has shown [9,44–46], *C. gloeosporioides* species complex (i.e., *C. fructicola*, *C. gloeosporioides*, *C. kahawae*, *C. viniferum*, *C. aenigma*, and *C. hebeiense*), *C. acutatum* species complex (i.e., *C. acutatum, C. limitticola*, *C. nymphaeae*, and *C. citri*), and *C. boninense* species complex (i.e., *C. karstii*) are the cause of grape anthracnose. In China, *C. viniferum* was reported as the dominant species in Beijing, Jiangsu, Shandong, Yunnan, Guizhou, and Fujian [9,44,47–49]. In this study, we collected diseased fruit from vineyards in four distinct geographical regions in Zhejiang Province. Through phylogenetic analysis and inoculation tests on grapes, we revealed that three *Colletotrichum* species, including *C. fructicola*, *C. aenigma*, and *C. pseudoacutatum*, cause grape anthracnose in Zhejiang. To the best of our knowledge, it is the first description of *C. pseudoacutatum* causing grape anthracnose.

*Colletotrichum* infects many cultivated fruit crops, such as grape, pear, citrus, strawberry, cherry, and banana [9,26,27,34,50,51]. Previous studies have shown that multiple *Colletotrichum* species or biotypes are associated with anthracnose disease on a single host [26,43,51,52]. Variation of the morphological traits of *Colletotrichum* species under different cultivation environments is common [23,26,27] and the morphology is not always reliable enough to distinguish *Colletotrichum* at species level. In our study, multi-locus phylogenetic analyses with concatenated ITS, *TUB2*, *ACT*, *CHS-I*, and *GAPDH* sequences revealed that *C. fructicola, C. aenigma,* and *C. pseudoacutatum* were associated with grape anthracnose in Zhejiang, and the morphological characteristics of each species also were described in detail.

In the current study, *C. fructicola* was the dominant species (40 isolates, 93%) on grapes, followed by *C. aenigma* (two isolates, 4.7%), and *C. pseudoacutatum* (one isolate, 2.3%). *Colletotrichum* bunch rot disease on mature grape fruit was first documented in the USA in 1891 [53]. First *C. gloeosporioides* (Penz) and then *C. acutatum* (Simmonds) have been reported to responsible for this disease [4,8,11], but the two *Colletotrichum* species were not detected in our study. *C. viniferum* has been reported as the main cause of grape anthracnose in some grape-planting areas in China [9,47–49]; however this species was not found in Zhejiang Province. Here, we hypothesize that grape variety affects the pathogen diversity of anthracnose [9,11]. Although *C. fructicola* was detected on *Colletotrichum*-diseased grapes at a low frequency in some vineyards in China [47–49], it was a key pathogen that caused anthracnose on grapes in all sampled regions in Zhejiang. In addition, *C. fructicola* is prevalent on pears and strawberries [26,34]. In this study, we provide the first record of *C. pseudoacutatum* as a cause of grape anthracnose, which is a newly recorded species that does not belong to the *C. acutatum* species complex [24].

Temperature is a major factor in plant fungal disease epidemics, and rainy, humid, and warm conditions, with temperatures ranging between 20 °C and 30 °C, are beneficial for *Colletotrichum*-disease epidemics [54–56]. Temperature affects the production and maturation of conidia of *Colletotrichum* spp., as well as its infection rate on grapes [4–6]. Mycelial growth of *Colletotrichum* spp. at different temperatures in vitro was determined, and our results were in agreement with a previous study that revealed the majority of *Colletotrichum* species grew at temperatures ranging from 10 °C to 35 °C [17]. In addition, our result showed that the mycelial growth of *C. fructicola, C. aenigma,* and *C. pseudoacutatum*

was inhibited at 5 °C and 40 °C, and they all had optimal growth temperatures on PDA around 25 °C, with *C. aenigma* having the lower growth rate.

Grape anthracnose is a polycyclic disease. Conidia are the source of infection by *Colletotrichum* species on grapes, and grape anthracnose epidemics are usually initiated by conidia, which settle on plant surfaces and germinate under favorable conditions and penetrate the host tissue [8,17]. Sporulation and the germination of conidia of the *Colletotrichum* species were determined, and the results showed that the highest sporulation was observed in two *C. aenigma* isolates; however, the conidial germination rate of *C. fructicola* was significantly higher than that of the other two species. Previous studies have shown that germination of conidia and sporulation are key aspects in the pathogen life cycle and are strongly correlated with infection of *Colletotrichum* species on hosts. The relative severity of disease caused by *Colletotrichum* species is determined by the proportion of conidia (conidia from overwintering primary sources or conidia produced on diseased tissues) that mature, disperse, and settle on plant tissue, then infect, and finally cause disease symptoms [6,17]. In our study, the disease index was used to represent the relative severity caused by *Colletotrichum* species on grapes. Consistent with the above studies, our results showed that the two *C. aenigma* isolates had poor conidial germination, resulting in lower pathogenicity.

## 5. Conclusions

In the present study, *C. fructicola*, *C. aenigma*, and *C. pseudoacutatum* were responsible for grape anthracnose in Zhejiang Province. According to our results, the *C. fructicola* species had the greatest spore germination rate and were the most pathogenic among the three *Colletotrichum* species, which may explain why it was the dominant species in vineyards in Zhejiang. *C. pseudoacutatum* was first recorded in grapes, and it also successfully infected strawberries and kiwifruit, indicating a lack of host specificity (data not shown). The lack of host specificity reminded us to focus on cross-infection caused by *Colletotrichum* species between grape and other horticultural crops. Benzimidazole resistance within the *gloeosporioides* complex on grapes was notable [57]. Here, we determined that *Colletotrichum* species caused grape anthracnose in Zhejiang vineyards, which will contribute to studying the epidemiology and fungicide sensitivity, and finally developing efficient methods for grape anthracnose management.

**Supplementary Materials:** The following supporting information can be downloaded at: https://www.mdpi.com/article/10.3390/agronomy13040952/s1, Table S1: List of *Colletotrichum* spp. strains used for multi-gene analysis in this study; Table S2: Colony diameter (mm) of *Colletotrichum* spp. under different temperatures (cultivation for 5 days on PDA); Table S3: Colony diameter (mm) of *Colletotrichum* species at 20 °C, 25 °C, 27 °C, and 30 °C (cultivation for 5 days on PDA).

**Author Contributions:** Conceptualization, J.W. and C.Z.; methodology, J.W., H.Y. and C.Z.; software, B.Y. and J.Z.; validation, H.Y. and C.Z.; formal analysis, J.W., J.Z. and X.C.; investigation, W.X. and X.C.; writing—original draft preparation, J.W. and B.Y.; writing—review and editing, H.Y. and C.Z.; visualization, J.W. and B.Y.; supervision, H.Y. and C.Z. All authors have read and agreed to the published version of the manuscript.

**Funding:** This research was funded by the National Innovation and Entrepreneurship Training Project for College Student (202110341052), sponsor: B.Y.

**Institutional Review Board Statement:** Not applicable.

**Informed Consent Statement:** Not applicable.

**Data Availability Statement:** Not applicable.

**Conflicts of Interest:** The authors declare no conflict of interest.

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
