# Peer review of "Genetic Diversity of Colletotrichum spp. Causing Grape Anthracnose in Zhejiang, China"

_agronomy, doi:10.3390/agronomy13040952_

Round 1

Reviewer 1 Report

Figure 3. Please elaborate the caption and describe the figure’s content.

For results subheadings like 3.1, Isolates and so on…please provide each heading with meaningful information from that particular result’s sub-section.

Figure 6 and 7 Captions are misplaced.

Authors may enrich literature on molecular markers (SSR/RAPD, etc). I suggest few references that could be useful. Multiplex molecular marker-assisted analysis of significant pathogens of cotton (Gossypium sp.), 2022; Biocatalysis and Agricultural Biotechnology https://doi.org/10.1016/j.bcab.2022.102557 ; Microsatellite analysis to differentiate clones of Thompson seedless grapevine, Upadhyay et al., 2010, Ind Journal of Horticulture, Volume 67 Issue 2 Pages 260-263.

Authors may consider to add conclusions section and provide future implications of the study in abstract as well as conclusions.

Author Response

We would like to thank you for your constructive comments concerning our manuscript (agronomy-2245701). These comments have helped to further improve the quality of our manuscript and clarify the significance of our research. We have carefully considered all of your comments and have made the necessary corrections. Our point-by-point responses to each comment are presented below.

Sincerely,

Jianyan Wu

Reviewer 2 Report

General comments

This paper enables to identify and characterize Colletotrichum species from vineyards in Zheijang in China. The study was conducted with a classical and adapted approach that enabled to evidence that Colletotrichum species from this region are different from other Chinese regions and C. pseudoacutatum was firstly described on grape.

The overall quality of the paper reaches the requirements but I listed some questions and corrections along the article:

Introduction

L35 : change in “km²”, change yield in “production”

L36: “is a serious disease”

L41: “is caused”

L47: “are observed”

L61: precise what you mean by “the epidemiology of gloeosprorioides differs from actutatum”

L62: “differs from”

L77: change to “heat and wetness favor grape anthracnose epidemics”

Material and methods:

2.1 Fungus isolation

Indicate the number of vineyards sampled in each region.

It should also be indicated how Colletotrichum isolates were differentiated if emerging from the same fruit. Were they identified based on spore shape? Or a designated number of germinated spores were cultivated in order to verify if different species were present in the same fruit?

2.5 Mycelial growth

L174: Change to “After 16h incubation”

2.6 Pathogenicity testing

L187: change to “were incubated in a growth chamber”

Were controls wounded too? If so it should be clarified

Results

3.1 Isolates (-> change the title to a more explicit title)

L206: change to “Grape berries showing anthracnose symptoms were sampled in vineyards from four…”

L208-210: simplify the sentence: “including 13 isolates from Hangzhou ….”

3.4 Multi-locus Phylogenetic Analyses

L214: change to “and 26 reference isolates”, “including C. boninense as an outgroup…”

L219: change to “40 isolates with … and 2 isolates with…”

L221: change like L214

Figure 2: the figure should be similar to figure 1 (form, font size…)

3.3 Species distribution

L241: change to “each sampled county”

L246: change to “only C. fruticola isolates were isolated from grape with anthracnose symptoms”

3.4 Morphology description

Make sentences in this paragraph for clearer understanding

Indicate the age of the colonies observed both in the text and Figure 4

L253: “The three C species…”

L255: “are described below”

3.5 Mycelial growth

Figure 7: repetitions should appear on Fig7a and the use of the same colors for each isolate will help to better identify the isolates among the 3 figures. Species names should be added in the legend for easier identification. No letters (statistics) are added on fig7b and c but described in the figure title.

Table 2: disease index of control should be indicated in the table (à size of the necrosis caused by the wound)

Discussion

The following papers should be cited:

-        Echeverrigaray et al 2020 (C. fruticola pathogenicity on grapes) “Colletotrichum species causing grape ripe rot disease in Vitis labrusca and V. vinifera varieties in the highlands of southern Brazil”

-        Lei et al 2016 “Identification and characterization of Colletotrichum species causing grape ripe rot in southern China”

-        Kim et al 2020 “First Report of Colletotrichum aenigma Causing Anthracnose of Grape (Vitis vinifera) in Korea

L329: change to “knowledge,…it is the first description of C. pseudoacutatum causing grape anthracnose”

L330: change to “Colletotrichum infects…”

L335: change to “to distinguish Colletotrichum at species level”

L343: change to “on grape mature fruits was first…”

L351: change to “in all sampled regions”

L384: change to “and were the most pathogenic among”

Author Response

(The authors gave the same response as above.)

Reviewer 3 Report

Dear Authors,

Material collected and examind in Your research is appropriate, and the results can be published in the Scientific Paper.

It could be good to check english grammar. F.ex. line 36 disease was? Sholud be is. Also line 41 and 42. Please check it.

Line 315 is there a chance to provide readers with the photographies of symptoms caused by all the mentioned in the article species? In the Article there are beautiful photographies of the colonies and spores of mentioned Fungi. It will be nice to attach also photos with the symptoms caused by Colletotrichum pseudoacutatum, C. fructicola, and C.aenigma on the grape berries.

Could Authors also mention in the discussion or in any other section section about chemical protection possibilities? Just a short sentence to provide the readers with the current state of chemical protection against Colletotrichum species in grape orchards.

Does the Authors planing to manage any tests with fungicides to examine to efficacy of fungal a.i. against mentioned species? There could be differences in the efficacy.

Reviewer

Author Response

(The authors gave the same response as above.)

Round 2

Reviewer 1 Report

The content is revised and updated in satisfactory manner by the authors.